# Does multidisciplinary videoconferencing between a head-and-neck cancer centre and its partner hospital add value to their patient care and decision-making? A mixed-method evaluation

Lidia S van Huizen [1,2] Pieter Dijkstra,[3] Gyorgy B Halmos,[4]
Johanna G M van den Hoek,[5] Klaas T van der Laan,[6] Oda B Wijers,[7] Kees Ahaus,[8]
Jan G A M de Visscher,[9,10] Jan Roodenburg[11]

Klaas T van der Laan died on May 4th 2019

For numbered affiliations see end of article.

**Correspondence to**
Lidia S van Huizen;
l.s.van.huizen@umcg.nl

## ABSTRACT

**Objectives** Given the difficulties in diagnosing and treating head-and-neck cancer, care is centralised in the Netherlands in eight head-and-neck cancer centres and six satellite regional hospitals as preferred partners. A requirement is that all patients of the partner should be discussed in a multidisciplinary team meeting (MDT) with the head-and-neck centre as part of a Dutch health policy rule. In this mixed-method study, we evaluate the value that the video-conferenced MDT adds to the MDTs in the care pathway, quantitative regarding recommendations given and qualitative in terms of benefits for the teams and the patient.

**Design** A sequential mixed-method study.

**Setting** One oncology centre and its partner in the Northern part of the Netherlands.

**Participants** Head-and-neck cancer specialists presenting patient cases during video-conferenced MDT over a period of 6 months. Semistructured interviews held with six medical specialists, three from the centre and three from the partner.

**Primary and secondary outcome measures** Percentage of cases in which recommendations were given on diagnostic and/or therapeutic plans during video-conferenced MDT.

**Results** In eight of the 336 patient cases presented (2%), specialists offered recommendations to the collaborating team (three given from centre to partner and five from partner to centre). Recommendations mainly consisted of alternative diagnostic modalities or treatment plans for a specific patient. Interviews revealed that specialists perceive added value in discussing complex cases because the other team offered a fresh perspective by hearing the case 'as new'. The teams recognise the importance of keeping their medical viewpoints aligned, but the requirement (that the partner should discuss all patients) was seen as outdated.

**Conclusions** The added value of the video-conferenced MDT is small considering patient care, but the specialists recognised that it is important to keep their medical

### Strengths and limitations of this study

► The study evaluates in depth the video-conferenced multidisciplinary team meeting (MDT) between the centre and the partner in the head-and-neck oncology care pathway and refocuses on benefits and drawbacks (strength).
► Participating specialists from different specialisms and locations were interviewed and identified benefits and drawbacks of the video conference meetings (strength).
► The researcher's presence during video-conferenced MDT may have influenced the communication between the centre and the partner, also called 'Hawthorne effect' (limitation).
► Only one of the six centres and its preferred partner in the Netherlands was studied (limitation).

viewpoints aligned and that their patients benefit from the discussions on complex cases.

## INTRODUCTION

Most tumours in the head or neck region (nasal cavity, paranasal sinuses, lips, mouth, salivary glands, throat or larynx and complex skin malignancies) are fast growing tumours.[1] This implies that a long interval between the moment of referral and the start of the primary treatment (surgery, radiotherapy and/or chemotherapy) can lead to tumour progression with less survival chance.[2] Because of complexity of diagnostic procedures and therapeutic modalities and low volume of patients, head-and-neck cancer care is centralised in multidisciplinary head-and-neck cancer centres.[3] In 1984, the Dutch Head & Neck Society (DHNS) was founded

as a scientific organisation. Later, the DHNS became involved in the nationwide organisation of head-and-neck cancer care. As a result, since 1993, patients of head-and-neck cancer in the Netherlands are treated in eight head-and-neck cancer centres recognised by the DHNS; six centres have preferred partners.[4] Within each head-and-neck cancer centre, multidisciplinary meetings according to national evidence-based guidelines are mandatory to provide the best diagnostic workup and treatment for patients and to sustain the quality of care in the oncology centres.[5–8] Criteria for qualifying as centre are: having the specialisms with expertise to treat the tumour, having the necessary diagnostic and therapeutic facilities and treating at least 200 new patients each year. Partners fulfil the same criteria but should treat at least 80 new patients.

In 1997, after an informal collaboration period of 4 years, the Medical Centre Leeuwarden became the formal preferred partner of the Head-and-Neck Cancer Centre of the University Medical Centre Groningen,[9] further referred to as the 'partner' and the 'centre'. The collaboration of a centre with its partner is based on trust and sustainable agreements on governance aspects, evidence-based multidisciplinary decision-making and use of facilities.[10–12] The collaboration consists of weekly multidisciplinary team meetings (MDTs) between centre and partner to discuss diagnostic and therapeutic plans. The efficiency of the MDTs is important for decision-making and care pathway management. The centre's MDT regarding diagnostics and treatment involves more than nine disciplines (details presented elsewhere).[13] The teams of centre and partner meet face-to-face three times a year, where governance, guidelines and research projects are discussed.

The DHNS and the Dutch Health Care Inspectorate (DHCI) require that all new patients of the partner are discussed in a weekly MDT with the centre.[14] This DHCI requirement can be seen as quality control over the partner clinic. Specialists from centre and partner, from the departments of oral and maxillofacial surgery (OMS), ear, nose and throat (ENT) and radiotherapy (RT) participate. This weekly MDT is additional to a local MDT in the hospital where the patient is first seen and will be treated. Initially, these collaborative multidisciplinary weekly meetings were in the centre: three specialists travelled to the oncology centre (2 hours travelling time and 2 hours MDT). When videoconferencing became available, it became the preferred method for this communication.[15 16] The video-conferenced MDT is scheduled after the local MDT. During the videoconferencing, the partner presents all patient cases, including available imaging, and proposed diagnostic and therapeutic plan. The centre presents complex cases or cases interesting to discuss. Both sides are free to offer recommendations. The team presenting the patient case is responsible for documenting changes when a recommendation is implemented.

Recommendations from both teams to the decision-making regarding diagnostic and therapeutic plans may add value to the quality of patient care.[17 18] We decided to evaluate the video-conferenced MDT as part of the collaboration agreements because it was time consuming and there was a wish to refocus on benefits and drawbacks.

### Research question

The aim of this study was to analyse the value of video-conferenced MDT in the treatment of head-and-neck cancer patients in the care pathways, resulting in two questions.
1. How often are recommendations given on diagnostic and/or therapeutic plans by the teams during video-conferenced MDT?
2. What benefits and drawbacks of the video conference are perceived by the specialists in the teams?

### DESIGN

This mixed-method study[19–21] had a quantitative part followed by a qualitative part. The primary outcome of the weekly video-conferenced MDT was the percentage of cases in which recommendations on diagnostic and/or treatment plans were given. The secondary outcome were the benefits or drawbacks of the MDT video conference perceived/experienced by the participating specialists. In the study period, the teams acted conform the DHCI requirement that all patients of the partner should be presented in a multidisciplinary meeting with the centre.

### Videoconferencing equipment used

The video-conferenced MDT was held in dedicated multidisciplinary meeting rooms, where screens can be operated with two to four computers with monitors. While the patient data are presented on the first screen, teams can see each other on the second screen. The videoconferencing is operated via the 'Webex' application and a camera. Both locations call into a special safe 'chat room'.

Centre: dedicated 20-seat videoconferencing room with three screens—beamers (software/provider Kinly; bandwidth 2 Mbps) and five camera inputs. Four computer stations, one dedicated for radiology showing Picture Archiving and Communication System (PACS) imaging.

Partner: dedicated 10-seat videoconferencing room with one screen with possibility to see patient data and the other team; one computer log-on to patient dossiers showing data and imaging.

### Patient data

Data of all patients presented by one of the teams during the video-conferenced MDT video conferences between September 2016 and February 2017 were included. The tumour localisation, histology and tumour stage were registered for all patients that were presented.

### Patient involvement in study design

Patients were not involved in the study because the main purpose of the study was to evaluate the added value of the DHCI requirement in a weekly video-conferenced MDT.

van Huizen LS, et al. BMJ Open 2019;9:e028609. doi:10.1136/bmjopen-2018-028609

**Table 1A** Definitions of change impact and case complexity: operational definitions of major and minor changes in diagnostic or treatment plan

|  | Diagnostic plan | Treatment plan | Remarks |
|---|---|---|---|
| Minor | Additional more-detailed MRI or CT thorax of the area already imaged | Logistic change | |
| Major | Additional MRI or CT thorax in a different area from the area already imaged | Change in modality: adding or deleting a therapeutic modality; surgery radiotherapy or chemotherapy | |
| Criterion | Addition of diagnostic plan in a different area than already investigated | Adding or deleting a treatment modality from the treatment plan in the proposed or in a different area | After the major/minor decision is made, the decision registered in the research form will be verified by both specialists (giver and receiver) |

**Table 1B** Definitions of change impact and case complexity: operational definition of case complexity

|  | Modality | Guideline | Comorbidity |
|---|---|---|---|
| Not complex | Unimodal treatment | Diagnosis and treatment follows guideline | No comorbidity |
| Complex | Multimodal treatment | Diagnosis and/or treatment does not follow guideline | Comorbidity |
| Remarks | ▶ Unimodal: surgical procedure chemotherapy primary radiotherapy ▶ Multimodal: reconstruction surgery chemoradiotherapy or bioradiotherapy | Which guidelines are followed | |

## Quantitative part

### Sample size calculation recommendations

In a 4-week pilot study of 4 sessions including 46 cases, carried out 9 months before study start, we found that in approximately 20% of cases a recommendation was given. To estimate this percentage with a 10% precision (95% CI: 15.5% to 25.4%) would require 250 cases. On average, 15 cases were discussed at each weekly video-conferenced MDT. We estimated that 6 months would be sufficient to acquire the necessary 250 cases. The pilot study was also used to operationalise the primary outcome measure.

### Recommendation registration

Recommendations were registered with the relevant data from electronic and written medical records on a clinical registration form by LSvH during the video conference. Each recommendation was assessed by the two teams with respect to change impact (minor or major; table 1A) on the diagnostic and/or therapeutic plan, case complexity, use of national multidisciplinary guidelines for the diagnostic and/or treatment plan and comorbidity of the patient (table 1B). LSvH registered the given recommendation with the relevant data; JGAMdV and JR verified the registrations. During the videoconferencing sessions, field notes were taken by LSvH.

### Statistical analysis

Differences in age, gender, tumour localisation and tumour histology (International Classification of Deseases in Oncology, ICD(O))[22] and tumour stage between cases presented by the centre and those presented by the partner were analysed using t-test for independent samples, $\chi^2$ test and $\chi^2$ test exact procedure if requirements for the $\chi^2$ test were not met. Statistical analyses were performed using SPSS V.23.0 for Windows software. In all analyses, statistical significance was set at the 5% level.

## Qualitative part

### Interviews

Semistructured interviews were conducted with six medical specialists that attended the meetings most frequently, one from the OMS, ENT and RT department of each team, to explore the added value of the video-conferenced MDT. The field notes taken by the researcher during the video-conferenced MDT were used to develop the questions for the semistructured interviews. After receiving verbal informed consent from the specialists, the semistructured interviews started with providing information about the recommendations given. Thereafter, it continued with the open question 'What do you think is the value of the video conference between the head-and-neck cancer centre and their preferred partner?'. LSvH then guided the interview using a short topic list including 'added value' and 'perceived possibilities for change or improvement in the video-conferenced MDT' (table 2). The different topics were introduced in a flexible way, and the interviews took the form of natural conversations.

Interviews lasted between 25 and 40 min, were audio recorded and transcripts of the interviews were made.

**Table 2** Interview guide

| Topics | Questions |
| --- | --- |
| Added-value videoconferencing | What do you think is the added value of the video-conferenced MDT between the head-and-neck cancer centre (centre) and their preferred partner (partner)? |
| | Could you mention strong points of the video-conferenced MDT? |
| | Could you give examples? |
| | Could you name points for improvement? |
| | Could you mention examples? |
| Role of specialism in video conference | What do you think the role of a specialist is in the video-conferenced MDT between centre and partner? |
| | The consultation is required by the Dutch Head and Neck Society and the Dutch Health Care Inspectorate, how usefulness do you think it is? |
| | Would you advise stopping the consultation if it was not mandatory? |
| Results interpretation | Have you given recommendations to the centre/partner? |
| | Have you received recommendations from the centre/partner? |
| | Could you indicate what the difference is between peer consultation and giving a recommendation? |
| | What do you think would be an ideal video-conferenced MDT? Could you explain your answer? |
| | What do you think could be adjusted in the video-conferenced MDT to make the consultation more effective and more efficient? |

The participants were asked to review the transcripts and extracted quotes, related to perceived added value, possible improvements and the role of a specialist in the video-conferenced MDT.

### Thematic analysis

Quotes were anonymised and coded for their relevance to possible benefits or drawbacks for the collaboration between the teams and for patient care. The first stage of this inductive analysis of the interviews involved two authors, JR and JGMvdH, in an initial open coding procedure that resulted in a list of codes corresponding closely to the text fragments extracted from the six interviews. The codes were placed in a coding tree using a thematic analysis approach with main themes recommendations, added value, collaboration and planning.[23 24] Codes were judged as being a benefit or a drawback. Any disagreements during the coding were discussed between the coders and the researcher.[25] After the preliminary results were collated, for credibility a member check was performed with participants.[26] The Clinical Research Office performed a planned quality check on data management.

## RESULTS

### Quantitative analysis

From September 2016 to February 2017, 82 patients were presented by the centre and 177 by the partner in 18 weekly video-conferenced MDTs (table 3). In this period of 22 weeks, three meetings were cancelled due to a 'medical complication meeting', a technical problem to connect and a holiday recess. Further, the researcher could not attend one session.

Most of the centre's patients (71 out of 82—86%) were presented only once, nine were presented twice (11%), one patient was discussed three times and another four times. Whereas 111 patients were presented only once (63%) by the partner. Generally, patients of the partner where presented twice or three times: the first time their diagnostic plan, the second time the therapeutic plan and sometimes surgical results the third time (55 out of 177—31%). Only one patient was discussed four times; five patients on the partner's list were not discussed at the first opportunity because imaging was not complete.

The partner presented significantly ($p<0.001$) more cases with infections that were initially suspected malignancy, T1-stage patients and non-complex cases. Tumour localisation and histology differed also significantly between centre and partner (table 3). In 61% of the 18 video conferences, both teams were complete; the centre team was not complete in 22% (n=4) and, in 17% (n=3), the partner team was not complete. On those occasions one of the other specialisms would present the cases, for example, OMS for ENT. The centre's ENT department was represented in most meetings by an ENT specialist training to be a head-and-neck oncology surgeon. The centre presented on average 5.2 (SD 2.4) cases per video conference, the partner presented on average 13.5 (SD 3.9) cases.

### Recommendations given

Recommendations were given in 8 of the 336 cases presented (2%; 95% CI: 1% to 5%) relating to 8 of the 259 patients (3%; 95% CI: 1% to 6%).

Of these recommendations, five were major and three minor (table 4). Four recommendations concerned diagnostic plans and four treatment plans. On three of the eight occasions when a recommendation was given, the centre's team was incomplete with one of the three specialisms absent. Seven of the eight recommendations

van Huizen LS, *et al. BMJ Open* 2019;**9**:e028609. doi:10.1136/bmjopen-2018-028609

**Table 3** Patients and their tumour characteristics, as presented during video conference meetings

| Number of patients (total n=259) | Centre (n=82) | | Partner (n=177) | | |
|---|---|---|---|---|---|
| (n=number of available data) | Mean | SD | Mean | SD | Statistics, p value |
| Age (mean, SD) | 67.8 | 15.2 | 66.7 | 16.1 | (t-test) 0.533 |
| Gender (n=259) | n | % | n | % | ($\chi^2$) 0.394 |
| Female | 27 | 10 | 68 | 26 | |
| Tumour localisation (n=206)* | n | % | n | % | ($\chi^2$ exact)<0.001 |
| Lip (C00) | 3 | 3 | 4 | 2 | |
| Oral cavity | 21 | 23 | 29 | 12 | |
| Tongue (C01, C02) | 6 | – | 11 | – | |
| Gums (C03) | 5 | – | 7 | – | |
| Floor of mouth (C04) | 4 | – | 4 | – | |
| Oral cavity, unspecified (C05, C06, C14) | 6 | – | 7 | – | |
| Major salivary glands (C07, C08) | 2 | 2 | 7 | 3 | |
| Oropharynx (C09,C10) | 7 | 8 | 6 | 2 | |
| Nasopharynx (C11) | 0 | 0 | 0 | 0 | |
| Nasal cavity (C30) | 2 | 2 | 3 | 1 | |
| Hypopharynx (C12, C13) | 5 | 5 | 5 | 2 | |
| Sinus (C31) | 3 | 3 | 3 | 1 | |
| Larynx (C32) | 10 | 11 | 15 | 6 | |
| Bronchus and lung (C34) | 0 | 0 | 5 | 2 | |
| Haematological and reticuloendothelial systems (C42) | 0 | 0 | 11 | 5 | |
| Skin (C44) | 14 | 15 | 35 | 14 | |
| Lymph nodes (C77) | 2 | 2 | 1 | 0 | |
| Unknown (C80) | 3 | 3 | 0 | 0 | |
| Miscellaneous (C20, 33, 41, 49, 50, 64, 73) | 3 | 3 | 7 | 3 | |
| Unknown (C80) | 3 | 3 | 0 | 0 | |
| Morphology or cell type (n=259) | n | % | n | % | ($\chi^2$)<0.001 |
| Squamous cell carcinoma | 57 | 72 | 78 | 44 | |
| Basic cell carcinoma | 3 | 4 | 6 | 3 | |
| Melanoma | 0 | 0 | 11 | 6 | |
| Miscellaneous malignant | 7 | 9 | 9 | 5 | |
| Benign | 2 | 2 | 18 | 10 | |
| Infection—premalignant abnormalities | 2 | 2 | 12 | 7 | |
| Miscellaneous | 11 | 13 | 43 | 24 | |
| T-stage (n=159)† | n | % | n | % | ($\chi^2$)<0.001 |
| T1 | 13 | 14 | 42 | 17 | |
| T2 | 20 | 22 | 20 | 8 | |
| T3 | 8 | 9 | 9 | 4 | |
| T4 | 25 | 27 | 14 | 6 | |
| Tx | 7 | 8 | 1 | 1 | |

In total 336 cases presented: 93 by centre and 243 by partner.

*Only tumour localisation if tumour diagnosed.

†Only TNM code if first diagnosed, so there are more patients in which 'localisation' is known (ie, for relapse or tumour residue or metastases).

were given by OMS specialists, and five of the eight were related to ENT patients. Seven of the eight instances occurred on a patient's first presentation and the other one during a second presentation although, in this case, the imaging had not been complete the first time. In general, recommendations were given related to the more complex cases, but not necessarily patients with comorbidity or those with more advanced tumours. About 70% of case were 'formalities' or 'routine patients', meaning patients that fitting the guidelines (well-defined tumours with limited regional metastases and without comorbidity).

**Table 4** Recommendation and its specifics

| No | Recommen-dation | Who | To whom | Team complete? | Recommen-dation (short) | Change impact, diagnosis or treatment phase | Patient status (ICD code, TNM classification, histology; case complexity, guideline used and comorbidity) | | | | | |
|----|-----------------|-----|---------|----------------|-------------------------|---------------------------------------------|------|------|-----------|----------|------------|-----------|
| | | | | | | | ICD | TNM | Histology | Complex? | Guideline? | Comorbid? |
| 1 | 2016G10-1 28-09-2016 | OMS partner | ENT centre | Yes | Give patient choice of expectative treatment | Major, treatment | C44 | T2N0M0 | SCC | Yes | No | Yes |
| 2 | 2016L14-1 28-09-2016 | OMS centre | OMS partner | Yes | Ultrasound-guided biopsy | Minor, diagnosis | – | – | Maligned lymphoma | No | Yes | No |
| 3 | 2016G32-1 26-10-2016 | OMS partner | OMS centre | Centre not | Use methotrexate to identfy malignancy | Minor, treatment | C00 | T1N0M0 | SCC | Yes | Yes | No |
| 4 | 2016G39-1 23-11-2016 | OMS partner | ENT centre | Yes | Change surgery approach to retain functionality | Major, treatment | C00 | T2N0M0 | Adenoid cystic carcinoma | Yes | No | No |
| 5 | 2016G40-1 23-11-2016 | OMS partner | ENT centre | Yes | Try PDT | Major, treatment | C01 | T4aN0M0 | SCC | Yes | No | No |
| 6 | 2016G51-1 14-12-2016 | OMS partner | ENT centre | Centre not | Consult ophthalmology | Major, diagnosis | C44 | T2N0M0 | BCC eye corner | Yes | No | Yes |
| 7 | 2016L90-2 14-12-2016 | OMS centre | ENT partner | Centre not | New biopsy | Major, diagnosis | C31 | T3NxM0 | Melan. | Yes | Yes | Yes |
| 8 | 2017L123-1 04-01-2017 | RT centre | OMS partner | Yes | Add MRI | Minor, diagnosis | C07 | T1N0M0 | SCC | Yes | Yes | No |

BCC, basal cell carcinoma; ENT, ear, nose and throat; ICD, International Classification of Diseases; Melan, melanoma; OMS, oral and maxillofacial surgery; PDT, photo dynamic therapy; RT, radiotherapy; SCC, squamous cell carcinoma.

van Huizen LS, *et al. BMJ Open* 2019;**9**:e028609. doi:10.1136/bmjopen-2018-028609

## Qualitative analysis—specialist interviews

During May 2017, six interviews were held. From the transcripts of the 6 interviews, 107 quotes were registered. During the coding procedure, items were placed in a coding tree with relevance to the primary research question (recommendations given) and the secondary research question (perceived benefits and drawbacks) by the researcher in consultation with the coders. For each major theme, minor themes were derived from the researcher's field notes. In total, 282 scores were given (table 5). In several instances, the quotes were scored differently although the intercoder agreement was acceptable given the possible 37 codes to choose from.

Benefits were more frequently mentioned by specialists of the partner, and the drawbacks more frequently by specialists of the centre. But the majority of codes had a positive connotation for the video-conferenced MDT (table 5).

Six main items were important according to the specialists (quotes in italic).

1. The video conference adds value when discussing complex cases, through assisting in fine tuning and aligning medical procedures (code 1, 20×).

*A patient is presented about which the own team had some discussion that can be discussed with the partner. In that manner, you get a confirmation or advice to change your treatment plan. This advice can be given by the same specialism but also by other members of the head-and-neck oncology team (ENT).*

2. Communication is essential for cooperation between teams (code 2, 10×); furthermore, it is important to know the partner well, not only via videoconferencing (code 13, 15×), and to interact respectfully (code 5, 10×) with mutual trust (code 7, 9×).

*The most important feature of the video-conferenced MDT is to communicate with each other on substantive medical matters, to be on speaking terms, and to know each other (RT).*

*During the videoconferencing, we respect each other, we listen to each other and we are open to each other's additional comments. We trust each other as partners (OMS).*

3. Recommendations are suggested alternatives on diagnostic modalities and treatment plans for specific patients (code 14, 17×).

*The video-conferenced MDT has the character of a collegial discussion, in which in collaboration the best diagnostic or treatment plan for your patient is reached. Confirmation on your treatment plan adds value too (OMS).*

4. For routine cases that fall within guideline for treatment, the video conference meeting adds little value as for changes in medical content, it can even irritate the participants in such cases (code 15, 9×).

The video-conferenced MDT sometimes changes the treatment plan for an individual patient. The video conference is not the meeting where new procedures or guidelines are developed (RT).

5. There is a wish to integrate the video conference with the site multidisciplinary meeting in both hospitals, the centre and the partner (code 17, 12×).

*Integration of the two local multidisciplinary meetings with the video-conferenced MDT could be valuable (ENT).*

6. The DHCI requirement (discuss all the partner's cases) has no added value. It is seen as old-fashioned or outdated (code 29, 8×).

*It is better to prepare at a high level and discuss, than to present all the patients and deal with each one briefly. Mutual preparation on special request could have added value, for example, a literature search on a complex osteosarcoma case (OMS).*

## DISCUSSION

Our results show that the added value of the weekly video-conferenced MDT between the head-and-neck cancer centre and the partner hospital was small given the few recommendations made on the initial diagnostic and/or treatment plan. Nevertheless, the specialists from both sites recognised the importance of keeping their medical viewpoints aligned through this type of communication. Whenever discussing complex cases in which a major change was recommended (in five of the eight recommendations), for example, to change the surgical approach to save functionality of organs or tissue, the recommended change in treatment had a large impact for that patient (table 4).

The data from the interviews suggest that especially complex patients would benefit from intercollegial consultation via video-conferenced MDT. If the teams were not obliged to discuss so many routine cases, they could use the time saved to prepare and discuss complex cases in greater depth.[27] The specialists said that they did not want to stop the video-conferenced MDT, because they appreciate reflecting on diagnostic and treatment plans with trusted expert colleagues.

Because of an increase in patients to be presented in the meeting, we were looking for a more efficient meeting, which could be reached not discussing the 'formalities' or 'routine patients' (about 70% of patients); developing an evidence-based working method would need more research. This result is supported by a large survey in the UK after 10 years of use of an MDT format, where specialists also said they wanted to change many components and refocus to spend more time on complex cases in detail.[18]

The qualitative part of this study showed that medical specialists perceived added value in discussing complex

**Table 5** Coding tree evaluation video-conferenced MDT

| Coding tree | | | | Pos? | Code | Code description | Partner | Centre | Total |
|---|---|---|---|---|---|---|---|---|---|
| Videoconferencing | Recommendation | Nuance | | + | 22 | Video-conferenced MDT is mostly 'intercollegial consultation' | 3 | 3 | 6 |
| | | | | + | 14 | Recommendations are nuances, not a totally different medical procedure or diagnostic/treatment plan for a specific patient | 7 | 10 | 17 |
| | | Follow-up traceable? | | + | 6 | Suggestions are taken from others | 1 | 2 | 3 |
| | | | | + | 20 | There is no patient-level evaluation on the implementation of medical procedures agreed, question of trust | 3 | 2 | 5 |
| | | | | – | 34 | Sometimes decisions are already taken in relation to continuity of treatment | 1 | 1 | 2 |
| | | Aligning | | + | 1 | Fine-tuning or aligning medical procedures | 10 | 10 | 20 |
| | | | | + | 9 | Continue routine cases discussion to prevent deviation from medical procedures | 2 | 2 | 4 |
| | | Knowledge | | 0 | 32 | Besides videoconferencing also bilateral consultation via telephone | 4 | 1 | 5 |
| | | | | + | 37 | Keep 'know how' with routine cases | 1 | 2 | 3 |
| | Added value? | Video-conferenced MDT | | + | 8 | Added value for complex cases versus routine cases | 21 | 24 | 45 |
| | | | | – | 15 | Little added value | 8 | 1 | 9 |
| | | | | 0 | 27 | Discuss radiotherapeutic scheme | 2 | 2 | 4 |
| | | | | – | 29 | Non-complex cases or 'formalities' are communicated because it is mandatory, no added value | 7 | 1 | 8 |
| | | | | + | 30 | Recommendation given to own discipline | 5 | 1 | 6 |
| | | Team completeness | | + | 4 | Presence of all three disciplines is essential | 3 | 4 | 7 |
| | | | | + | 11 | Expertise (good) of physician is important | 5 | 3 | 8 |
| | | | | 0 | 23 | Add presence of medical oncology discipline as expertise | 2 | 2 | 4 |
| Collaboration | | Communication | | 0 | 2 | Working together requires communication | 8 | 2 | 10 |
| | | | | + | 10 | At both locations working methods are comparable through video-conferenced MDT | 5 | 2 | 7 |
| | | | | – | 19 | Initially it was good to consult, added value decreased because teams have grown towards each other | 1 | 1 | 2 |
| | | Trust | | + | 5 | Respectful collaboration | 3 | 7 | 10 |
| | | | | + | 7 | Mutual trust | 4 | 5 | 9 |
| | | | | + | 13 | Important to know the partner, not only via videoconferencing; good for cohesion | 8 | 7 | 15 |
| | | Expertise | | – | 18 | Centre member does not think videoconferencing necessary, because partner should be trusted as such | 2 | 4 | 6 |
| | | | | + | 26 | Expertise and new developments from centre to partner | 2 | 2 | 4 |
| | | DHCI requirement | | 0 | 21 | Video-conferenced MDT between centre and partner is a national agreement or policy | 2 | 3 | 5 |
| | | | | – | 31 | The national policy—to discuss all cases including routine cases—between centre and partner is perceived as outdated | 7 | 2 | 9 |

Continued

van Huizen LS, *et al. BMJ Open* 2019;**9**:e028609. doi:10.1136/bmjopen-2018-028609

**Table 5** Continued

| Coding tree | | Pos? | Code | Code description | Partner | Centre | Total |
|---|---|---|---|---|---|---|---|
| Planning | Logistics | – | 16 | Stressful, considering other video conferences | 3 | 6 | 9 |
| | | 0 | 17 | Integrate video-conferenced MDT in the hospital's MDT for centre and partner | 5 | 7 | 12 |
| | Preparation | – | 12 | Improve format of patient presentation | 1 | 1 | 2 |
| | | + | 24 | Good preparation is important | 5 | 4 | 9 |
| | Commitments | + | 25 | Starting and stopping the video-conferenced MDT on time is important | 4 | 1 | 5 |
| | | 0 | 33 | Possibly cancel video-conferenced MDT when nothing to discuss | 1 | 1 | 2 |
| | Equipment | + | 3 | Technique always flawless | 1 | 1 | 2 |
| | | – | 35 | Sometimes video-conferenced MDT did not take place due to technical malfunction | 1 | 1 | 2 |
| | | – | 36 | Placement of monitor in the room hinders colleagues and hampers interaction | 2 | 2 | 4 |
| Scientific research | | 0 | 28 | Bias through research setting because researcher is present as observer (Hawthorne effect) | 1 | 1 | 2 |
| Total quotes | | | | | 151 | 131 | 282 |

This coding tree has major and minor themes that were derived from the primary research question (recommendations given), the secondary research question (added value as described in benefits and drawbacks perceived) and minor themes derived from researcher's field notes. One code was related to the research situation.
'Pos?' refers to the question: has this code a positive connotation or benefit? + = yes, 185 scores; 0 = neither positive nor negative, 42 scores; – = no, 55 scores.
The amount of codes given is given for the partner, the centre and in total.
DHCI, Dutch Health Care Inspectorate; MDT, multidisciplinary team meeting.

cases in a collegiate consultation, because the other team offers a fresh perspective by hearing the case 'as new'. Although remarks were often about nuances, the confirmation on the chosen treatment by the other team was experienced as helpful. This view is supported in literature where medical specialists found videoconferencing useful in at least one aspect of their practice.[10]

An important requirement to communicate through video conference is that participants know each other from personal meetings to support mutual trust and respect as the basis for cooperation. The finding that collaboration and cooperation improves when each discipline understands each other's roles and that specialties working together for a long time do not need many words to come to a decision was supported previously.[17 28]

The video-conferenced MDT can be used to introduce and discuss new developments, protocols and guidelines leading to comparable quality of care in both locations. Comprehensive cancer centre teams working together over videoconferencing with a peripheral hospital team, reviewing radiotherapy planning align their treatment plans (7% major and 21% minor changes)[16] and speed up follow-up appointments.[15]

The video-conferenced MDT differs from the local MDT: complex cases are discussed with a second 'expert team' of head-and-neck oncology specialists. The patients treated by the centre and partner are similar, although diagnostics and treatment might differ slightly,[29] only in case of rare tumours that need skull base surgery patients travel from partner to centre. In our study, the significant differences in tumour localisation, cell type and tumour stage between sites are a consequence of 'the DHCI requirement', whereas the 'centre' could decide which of its patients would make an interesting case for discussion. Consequently, the partner presents three to four times as many patients as the centre. One-third of these (31%) reappeared in the subsequent video conferences, checking extra diagnostic information, treatment plan and need for adjuvant therapy. Most of these presentations were seen as a 'formality'.

The perceived value of the video-conferenced MDT might be influenced by the expertise of specialists. The recommendations given during the evaluation period were mostly given to ENT by an OMS oncologist who had considerably more clinical experience than his opposing colleague had, and was one of the instigators of the collaboration. It could be that recommendations given were accepted more easily if given by a more experienced specialist.[12] Videoconferencing enables specialists acquiring experience with presenting patients with complex oncology and with decision-making in teams.[6 17]

### Limitations of this study
Contrary to our findings from the 4-week pilot study (n=46), where advice was offered in 20% of the presented cases, the actual 2% recommendations is much lower. Although it is difficult to explain this difference in amount of 'agreed recommendations', we think that the

pilot served mainly as a feasibility check that helped us to define our research questions and to operationalise the definitions. Other factors may also have played a role in the difference between the pilot and the actual study. First, the long-lasting collaboration between the centre and the partner had led to a high level of alignment on diagnostic and therapeutic 'strategies' or medical viewpoints. Second, the participants were not blinded for the research question. Thus, awareness of being part of an experiment may have led to a drive to perform well and to present the patients with an optimal diagnostic and treatment plan (Hawthorne effect). Additionally, the presence of the researcher might have influenced the communication between centre and partner. Often, the teams mentioned that the other team was asked to give collegial advice, and therefore, a suggestion was not always seen as a recommendation. This nuance could also be interpreted as a social desirable answer, possibly due to the long existing collaboration between the centre and the partner before study start. Third, some patient cases were only presented as interesting to discuss. Finally, during the pilot study, the advice given was not assessed for its impact.

In this study, we evaluated the added value of a video-conferenced MDT between one oncology centre and its preferred partner. In line with other studies,[30 31] this study showed that, in addition to a quantitative result (number of recommendations), it is important to reflect on the situation through an interview process (qualitative results) before starting to implement improvements. The interviews showed that specialists from both centre and partner support the idea of sustainable collaboration, but they do not support the view implicit in the DHCI requirement that the centre should act as means of quality control for the partner.[32] Our findings on video-conferenced MDTs find support elsewhere in terms of the positive results on teams working together.[33–35] Other studies have shown that more research is needed to understand the effects of video-conferenced MDT on patient outcomes, such as finance including resource usage,[36 37] what fields of specialisms could benefit from the medium,[28 38] participant satisfaction,[39] throughput times[40] and self-management for patients.[41]

In summary, we believe that the DHCI requirement (the partner should discuss all patients with the centre) is unnecessary in the case of routine patients, since it does not add value to the quality of their treatment. It is more useful to spend time to discuss complex cases in greater detail. We propose the following measures that will add value to the weekly video-conferenced MDT:

1. All the participating medical specialists should be granted freedom to select only complex or interesting cases that could serve to keep medical procedures aligned.
2. The partner should not be obliged to present cases seen as 'routine patients' since this does not add value.

3. The video-conferenced MDT should be organised as an integral part of the partners' MDT and not as a separate weekly meeting.
4. Accepted, mature processes should be regularly reassessed and refocused to enable new collaboration strategies.

Based on our findings on the added value of the multi-disciplinary video conference between the head-and-neck centre and its partner and our suggestions for improvements, we would advise the DHNS, along with healthcare policymakers, to reconsider the DHCI requirement.

In our study, we found that there are clinical and practical implications on how and when to start with videoconferencing instead of meetings with physical attendance. Videoconferencing must be seen as a supportive medium for communication within a sustainable collaboration of parties that understand each other's roles and work with guidelines or protocols.

Participants of a video conference should:
1. Know each other and meet face-to-face on a regular basis, which serve cohesion (management meetings on governance, guideline developments and research projects are ideal for this purpose).
2. Respect each other as 'expert/knowing' colleague and know each other's role in the multidisciplinary treatment of patients.
3. Trust each other in follow-up of changes to diagnostic and treatment plans.

In view of the above-mentioned implications, we would not recommend starting with videoconferencing for multidisciplinary meetings if a majority of participants do not know each other.

## CONCLUSIONS

The video-conferenced MDT has added value in the collaboration and in the care pathway management. When interpreting national multidisciplinary guidelines, centre and partner align their medical policies. This leads to a more efficient use of resources and work force.

Conversely, discussing non-complex cases is seen as a burden and the DHCI requirement to discuss all the partners' cases as outdated.

**Author affiliations**
[1]Quality and Patient Safety, University of Groningen, University Medical Center Groningen, Groningen, The Netherlands
[2]Oral and Maxillofacial Surgery, University of Groningen, University Medical Center Groningen, Groningen, The Netherlands
[3]Centre for Rehabilitation, University of Groningen, University Medical Center Groningen, Groningen, The Netherlands
[4]Ear, Nose and Throat, University of Groningen, University Medical Center Groningen, Groningen, The Netherlands
[5]Radiotherapy, University of Groningen, University Medical Center Groningen, Groningen, The Netherlands
[6]Ear, Nose and Throat, Medical Center Leeuwarden, Leeuwarden, The Netherlands
[7]Radiotherapeutic Institute Friesland, Leeuwarden, The Netherlands
[8]Erasmus School of Health Policy & Management, Erasmus Universiteit Rotterdam, Rotterdam, The Netherlands
[9]Oral and Maxillofacial Surgery, Medical Center Leeuwarden, Leeuwarden, The Netherlands

¹⁰Oral and Maxillofacial Surgery/Oral Pathology, Free University Medical Center, Amsterdam, The Netherlands
¹¹Oral and Maxillofacial Surgery, University of Groningen, University Medical Center Groningen, Groningen, The Netherlands

**Acknowledgements** This research was sponsored by the University Medical Centre Groningen.

**Contributors** LSvH was involved in the study design and concept, carried out the study, performed the statistical analysis and the analysis and interpretation of the data and drafted the manuscript. PD, KA, JGAMdV and JR, the supervisor, were involved in the study design and concept, analysis and interpretation of the data, and revision of the manuscript. JGAMdV and JR were involved in the coding of the interview quotations, together with LSvH. GBH, JGMvdH, KTvdL and ODW were involved in the acquisition of the data and the revision of the manuscript. All authors read and approved the final manuscript.

**Funding** The authors have not declared a specific grant for this research from any funding agency in the public, commercial or not-for-profit sectors.

**Competing interests** None declared.

**Patient consent for publication** Not required.

**Ethics approval** This prospective observational study on decision-making analysis was checked by the Medical Ethics Review Board of the UMCG (2016, ref. M16.194909), the Netherlands. They concluded that the study is not a 'clinical research study with human subjects' as meant in the Medical Research Involving Human Subject Act (WMO). The Dutch law requires also a privacy statement from the partner in the study, the Medical Centre Leeuwarden (2016, nWMO 187).

**Provenance and peer review** Not commissioned; externally peer reviewed.

**Data availability statement** Data are available upon reasonable request.

**Author note** The University Medical Center Groningen is developing patient-centred care pathways for diverse patient groups including laws and regulations for quality and patient safety. LSvH and JR are working in cooperation with KA to research care pathway implementation in the Comprehensive Cancer Center Groningen and to develop quality and safety indicators, that is, process indicators that predict performance of care pathways and sustainable patient outcome.

**ORCID iD**
Lidia S van Huizen http://orcid.org/0000-0002-8106-4190

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
