## [Reviewer comments · BMJ Open]

ARTICLE DETAILS

TITLE (PROVISIONAL)	Does multidisciplinary videoconferencing between a head and neck cancer centre and its partner hospital add value to their patient care and decision making? A mixed method evaluation
AUTHORS	van Huizen, Lidia; Dijkstra, Pieter; Halmos, Gyorgy; van den Hoek, Johanna; van der Laan, Klaas; Wijers, Oda; Ahaus, Kees; de Visscher, Jan; Roodenburg, Jan

VERSION 1 – REVIEW

REVIEWER	Tayana Soukup King's College London London, UK
REVIEW RETURNED	06-Mar-2019

GENERAL COMMENTS	I would like to commend the authors on their work. This is a very well conducted study and a clearly written and structured paper. The only thing I would recommend is that the authors reflect on the limitations of their study and add this information into their discussion under the subheading 'Limitations' - this will greatly strengthen the paper.
---

REVIEWER	James Green Barts Health NHs Trust London UK
REVIEW RETURNED	04-Apr-2019

GENERAL COMMENTS	The impression of the interviews was that the interviewees seemed to think video conferencing was a success as it had built trust and collaboration, and aligned 'thoughts' on care. The study shows either very high levels of alignment and agreement (which was not investigated per se) or could show very low levels of change. In many parts of the world the MDT meeting has a dual role; a governance aspect and to facilitate evidence based decision making on the options of care for patient discussed. The article gives the impression that the pairing of centres was set up in the Netherlands for similar reasons but it is not categorically said that these are the reasons that the MDT was set up. What do the authors think are the overarching reasons that this MDT model was set up in the Netherlands for head and neck. Following on from this it might be useful to explain a little more to readers who do not understand the organizational reasoning behind the centre/partner model to drivers for the formation of the MDT.
--

	Was there a problem before this time that the MDT model was trying to solve. Lack of standardisation of practice Lack of specialist staff Poor access to a specific treatment modality Etc etc Could there be more detail of what criteria makes a centre chosen to be the centre and why a partner is deemed the partner ? Is it due to having superior treatment modalities or access to different chemo therapy / oncology or specialist support. ? line 322 'although diagnostics and treatment might differ slightly'. Do patients ever travel from the partner site to the centre to access higher orders of treatment ? That was not the impression given in the article so clarifying this is important. If they do travel between hospitals. An idea of how many patient discussed by the partner site then go onto received treatment locally and how many receive treatment at the centre would be useful. Especially if the percentage moving to the centre for treatment is similar in proportion to the number needing full MDT discussion. If the percentage is not comparable. The accepted guidelines may be highly robust and a discussion point on this may be useful. It is encouraging to see the team in the Netherlands re-assessing the value of a videoconference MDT once it had 'matured'. This is sensible and timely as it is all too easy to continue with the status quo. It may be sensible to make that point more strongly to encourage others to regularly reassess accepted processes, that were set up many years ago to solve a problem in healthcare, but had outgrown their usefulness. In the UK a large survey by Cancer Research UK assessed cancer clinicians views on MDTs after 10 years of their use and found teams (including head and neck teams) wanted to change many components and refocus the reasons for discussion . Likewise Cancer teams in the UK felt 'times had moved on' and many of the reasons MDTs had been set up (and Operating guidance) were outdated because pathways were so developed i.e. had matured to such a level, 10 years after their inception, that discussion of every patient was no longer needed. In a similar way to this study. I think in the discussion it would be helpful to make this point - perhaps quoting/referencing this work (line 332) https://www.cancerresearchuk.org/sites/default/files/full_report_meeting_patients_needs_improving_the_effectiveness_of_multidisciplinary_team_meetings_.pdf It would good if the authors postulated further on why such a marked difference occurred in percentage change between the pilot and the fuller study, as it was large . Was it due to the Hawthorn effect ? Did anyone ask the teams why they thought this occurred and which percentage was the nearer the mark 2% or 20%, in the opinion of the teams. Was the pilot carried out when the return to videoconferencing was quite new if so a run chart or a control chart might have been a better way of assessing the change over time as the MDT 'matured' ? As the amount of change was very low It would be interesting for the researchers to calculate
--	--

	a) how many patients did NOT need a discussion due to robust clinical guidelines b) an estimate of how many needed to be brought the videoconference joint MDT using their proposals and how much resource this could save (clinicians time etc). Is there any costing for MDT time in the Netherlands ? c) Is there was a robust way of identifying who needed to be brought for a full discussion – what were the common criteria (even if the small numbers means this is not possible it should be included as a focus for further research) ? As the authors are saying that not all cases should be discussed, a guideline on who should be discussed (the category of patient/disease/complexity) would probably be useful going forward. Or it may be easier to say categorically who should NOT be discussed. I realize the authors are saying 'not all new cases' but surely some may necessitate discussion. If so why ? Line 355 proposed to discuss more complex cases - supports the finding of CRUK study which also assessed head and neck cancers (see reference above) -MDTs are thought to save time elsewhere administratively and due to familiarity with the case after presentation. Was there any data to show that staff felt the discussion saved them time/resource elsewhere in the patient care pathway ? -Staff seemed to identify other organizational factors that they would like to improve, in the discussion could solutions to these issues be proposed.. The whole team (centre and partner) could have a face to face regularly/annually so that guidelines , cross sites improvement and trials can be proposed, discussed and implemented , -There could be more discussion of other MDT models instead of video conferencing (perhaps a virtual model, if the number of cases are so small, the cases could be perhaps put up on a website 1 week before and specialists comment on the best care) that could provide a solution to the resources needed to run this MDT. -who else is present in the Head and neck MDT as in the UK CNSs and therapists are present, is this the cases in the Netherlands ? They often think their time is saved later (see above) as they are familiar with the case. Perhaps stating the team that makes up the MDT in each site, under standard guidance/minimum standard expected would be useful. If they are present why were only doctors interviewed ? -Could the authors please state on which sites the 6 specialists worked who were interviewed in the qualitative aspect of the study. Were 3 individuals at the centre and 3 at the partner site? Could their specialities be given and had all finished training i.e. were they all qualified staff -There appeared to be medical trainees presenting cases, why was this, were people on leave ? Would one expect fully trained staff at each site in the model/operating guidance to be doing so ? -I couldn't understand why in table 2 Gender was included as a mean (with SD) when absolute numbers might have been more useful
--	---

	-Attendance seems to be an issue and if full attendance had occurred at the centre then 4 recommendations (proposed by the partner site) the might have been removed (leaving 1%) - is cross covering for leave etc a planned function of the centre/partner MDT model, if so can this fact be including in the discussion/introduction. As if so the 2% may not necessary be due to 'change' but due to cross-covering a speciality by one site when others are away. -346 enjoy team working - There was little mention of other valuable aspects that videoconferenced MDT / access to supra-specialist advice can bring : Education of teams/juniors, discussion of whether patients might be eligible for Research trials etc. Where any of these other aspects brought up by the interviewees as important aspects of the meeting ? where they asked this ? It would perhaps be useful to explain these other positive aspects of MDTs to readers who may be unfamiliar with MDTs . -Just meeting over videoconferencing is thought to be suboptimal for team working. Moving from a face to face meeting in one room, to videoconferencing it would be sensible to put more social/team interaction in place to facilitate improvements and development of services . This is often unappreciated so a paragraph about this would be useful.
--	--

REVIEWER	Joan Prades Catalan cancer plan, Health department, Catalonia, Spain & University of Barcelona (UB)
REVIEW RETURNED	14-Jun-2019

GENERAL COMMENTS	In the field of oncology, the head and neck cancer is typically known by the conflictual decision-making process given the existing different therapeutic alternatives and the radical impact of these in terms of short, secondary and secondary effects. On the other hand, the introduction of ICTs in cancer care is a main challenge. Both issues make this article relevant. Nevertheless, I would like to make some remarks. Major points I consider the organisation of the Methodology section quite weak. On one side, it is essential to stress how the qualitative strand is "mixed" with the quantitative strand. Just as an option, I would advise the authors using the classical reference of "Creswell JW and Plano Clark VL. Designing and conducting mixed methods research. USA: Sage Publications; 2011" in order to better clarify how you organised the whole study. It is of great importance to make explicit elements such as how both strands relate and were mixed. For instance, was that a convergent or a sequential design? Was the design explorative or explanative? That is, were the qualitative results intended to explain the quantitative ones? And, also, were the results from both strands mixed at the study design stage, or were they mixed at the Results or Discussion stages? This should be introduced in the "design" sub-section. A second element that can be stressed is the lack of order in the presentation of both strands. From page 5 to page 8 quantitative and qualitative elements (patient data, qualitative analysis, sample size...) are shown without splitting the quantitative study from the qualitative study. After giving more details about the study design,
---

I would separate the methodological aspects of the quantitative assessment (or evaluation or sub-study) from the qualitative ones. Other key points are the following. It was not explained how the interviewees were selected. It seems obvious but a comment could be introduced as long as the sample of informants is quite small and there may be a risk for respondents' biases such as "social desirability".

Just as an opinion, I do not consider very relevant the distinction between "recommendation" and "suggestion" (p 17, line 300), being the latter rather "induced". The point is whether the clinical discussion impacted the clinical management of patients, ultimately.

I'm not totally sure of these two points—page 3, line 63 and 66—being strengths of the study. They seem more comments about the results. I think they have nothing to do with the design of the study or the methods used. Likewise, another key limitation should be added: the results of the study can be biased due to the 4-year pilot carried out previously. This is described by the authors. It could be considered a limitation in view of the low percentages of change in the recommendations.

I would like to acknowledge the work done and the commitment to transparency in showing the coding tree evaluation. It has been really interesting reading the codes' description.

Minor points

Three other minor aspects observed in the Methodology section were the following. In page 5, line 131, it is explained that patient data were included but table 2 is not mentioned. Also, in page 6, it is stressed why patients were not involved. If they were not the target, then is not necessary to include a specific sub-section justifying it. Finally, we do not know the specific type of qualitative analysis performed. Considering the codification process and the inductive way used to create contents, I presume that it corresponds to the thematic analysis.

Concerning the Abstract section, I find it difficult to understand therein the meaning of "Collaborating contract" without reading the whole paper (page 2, line 37). I would rephrase it by saying "as part of a regional health policy rule" or something like that. Also, it is mentioned in the "objectives" that this is a mixed methods study while the "Design"/Methodology section do not contain this information. This should be changed.

Other points

It is used the word "policy" to mean that you have a treatment plan for a patient (e.g. line 306). Personally, I would not use in this paper the word policy in that sense.

Although this comment goes probably beyond the scope of a reviewer, I wonder about the category "routine patient" mentioned on page 19, line 354. From my experience, for some physicians a routine patient is someone with a low degree of cancer-related clinical complexity, but this is excluding patients with some comorbidity, patient preferences which are not those of clinicians, etc. Reaching a minimum consensus on the concept of "routine patient" could be interesting. It might help operationalising the avoidance of discussions on these patients (and the feeling of wasted time) while there are no patients who deserve to be discussed by both teams and instead they are not.

The idea of efficiency is not included in the text. The results of this study are coherent with the idea of providing high-quality care (by promoting horizontal integration between teams from different

	hospitals) while making this change sustainable. Just an option, this could be included in the conclusions.
--	---

VERSION 1 – AUTHOR RESPONSE

Reviewers' comments and rebuttal 'Does multidisciplinary videoconferencing between a head-and-neck cancer centre and its partner hospital add value to patient care and decision-making? A mixed method evaluation'

Reviewer(s)' Comments to Author:

Reviewer: 1 Tayana Soukup, Institution and Country: King's College London, London, UK

This is a very well conducted study and a clearly written and structured paper.

Authors reply

Dear reviewer, dear Tayana, thank you for your compliment on the conduct of our study and the structure of our paper.

We reflected on your comment in the table below.

no	reviewer comment	reply author	changes in document
1	Reflect on the limitations of the study and add this information into the discussion under the subheading 'Limitations' - this will greatly strengthen the paper.	We agree this is helpful, see also comment (reviewer 3, comment 4 and 5)	We added in the discussion, page 19, line 347 the subheading 'limitations of this study'.

Reviewer: 2, James Green, Institution and Country: Barts Health NHs Trust, London, UK

The impression of the interviews was that the interviewees seemed to think video conferencing was a success as it had built trust and collaboration, and aligned 'thoughts' on care.

The study shows either very high levels of alignment and agreement (which was not investigated per se) or could show very low levels of change.

In many parts of the world the MDT meeting has a dual role; a governance aspect and to facilitate evidence based decision making on the options of care for patient discussed.

The article gives the impression that the pairing of centres was set up in the Netherlands for similar reasons but it is not categorically said that these are the reasons that the MDT was set up.

Authors reply

Dear reviewer, dear James, thank you for interesting comments on the outcomes of our study and the connection to research that was done in the UK on oncological care and MDT development, with our paper.

We reflected on your comment in the table below.

no	reviewer comment James Green	reply author	changes in document
1	What do the authors think are the overarching reasons that this MDT model was set up in the Netherlands for head and neck. Following on from this it might be useful to explain a little more to readers who do not understand the organizational reasoning behind the centre/partner model to drivers for the formation of the MDT.	In 1993 concentration of care due to high complex – low volume and involvement of many disciplines in H&N cancer care, it was decided to concentrate the care in H&N Oncology Centres.	To explain why the additional MDT was by VC we added a sentence in the introduction, page 5, lines 113-114: Initially, these collaborative multidisciplinary weekly meetings were in the centre: three specialists

no	reviewer comment James Green	reply author	changes in document
	Was there a problem before this time that the MDT model was trying to solve.  Lack of standardisation of practice Lack of specialist staff Poor access to a specific treatment modality Etc etc 	The video conferenced MDT between the centre and the partner is an additional MDT to their own, to align procedures of the centre and the partner as a 'quality control measure', and is called 'the DHCI requirement' in the paper.	travelled to the oncology centre (2 hours traveling time and 2 hours MDT).
2	Could there be more detail of what criteria makes a centre chosen to be the centre and why a partner is deemed the partner? Is it due to having superior treatment modalities or access to different chemo therapy / oncology or specialist support?	Criteria for the choice of hospital to become an oncology centre:  1. All necessary specialist disciplines present 2. Diagnostic and therapeutic Facilities 3. At least 200 new patients each year for the centre and 80 for the partner. These criteria are published in the Dutch SONCOS standardization document (Stichting Oncologische Samenwerking – the Dutch Foundation on Oncological Cooperation) for all tumour types treated, that is updated each year.	To explain we added a sentence to introduction, page 4, lines 95-97: Criteria for qualifying as centre are having the specialisms with expertise to treat the tumour, having the necessary diagnostic and therapeutic facilities and treating at least 200 new patients each year. Partners fulfil the same criteria, but should treat at least 80 new patients.
3	line 322 'although diagnostics and treatment might differ slightly'. Do patients ever travel from the partner site to the centre to access higher orders of treatment? That was not the impression given in the article so clarifying this is important. If they do travel between hospitals. An idea of how many patient discussed by the partner site then go onto received treatment locally and how many receive treatment at the centre would be useful. Especially if the percentage moving to the centre for treatment is similar in proportion to the number needing full MDT discussion. If the percentage is not comparable. The accepted guidelines may be highly robust and a discussion point on this may be useful.	Referral from partner to the centre is rare, due to the fact that the partner meets the same criteria as the centre except for the amount of patients treated. In the research period 1 patient out of 177 was referred to the centre by the partner.	We added in the discussion page 18, line 332: 'The patients treated by the centre and partner are similar, although diagnostics and treatment might differ slightly; only in case of rare tumours that need skull base surgery patients travel from partner to centre.'
4	It is encouraging to see the team in the Netherlands re-assessing the value of a videoconference MDT once it had 'matured'. This is sensible and timely as it is all too easy to continue with the status quo. It may be sensible to make that point more strongly to encourage others to regularly reassess accepted processes, that were set up many years ago to solve a problem in	Thank you for your appreciation for evaluating the value of videoconferencing MDT. Although videoconferencing is now a widely accepted communication tool, the video conferenced MDT between the centre and the partner needs to	We added a sentence in the introduction, page 5, lines 122-124: We decided to evaluate the video-conferenced MDT as part of the collaboration agreements because it was time

no	reviewer comment James Green	reply author	changes in document
	healthcare, but had outgrown their usefulness.	be evaluated on its usefulness, because it is also time consuming and is an additional MDT to the sites own MDT. In the paper we called it 'the DHCI requirement'. To explain why the additional MDT was by VC we added a sentence (see comment 1).	consuming and there was a wish to refocus on benefits and drawbacks.
5	In the UK a large survey by Cancer Research UK assessed cancer clinicians views on MDTs after 10 years of their use and found teams (including head and neck teams) wanted to change many components and refocus the reasons for discussion. Likewise Cancer teams in the UK felt 'times had moved on' and many of the reasons MDTs had been set up (and Operating guidance) were outdated because pathways were so developed i.e. had matured to such a level, 10 years after their inception, that discussion of every patient was no longer needed. In a similar way to this study. I think in the discussion it would be helpful to make this point - perhaps quoting/referencing this work (line 332) https://www.cancerresearchuk.org/sites/default/files/full_report_meeting_patients_needs_improving_the_effectiveness_of_multidisciplinary_team_meetings_.pdf (report 2017).	Your input has helped us in clarifying the evaluation of the care path way. We referred to the CR UK report in the introduction (2017), we used it in the discussion to underpin our findings and added in the last part of the discussion an additional 'measure'.	We added the reference in introduction, page 5, line 122. We added in the discussion, page 20, line 386, a 4th measure: Accepted, mature processes should be regularly reassessed and refocused in order to enable new collaboration strategies.
6	It would be good if the authors postulated further on why such a marked difference occurred in percentage change between the pilot and the fuller study, as it was large. Was it due to the Hawthorn effect (line 299)? Did anyone ask the teams why they thought this occurred and which percentage was the nearer the mark 2% or 20%, in the opinion of the teams. Was the pilot carried out when the return to videoconferencing was quite new if so a run chart or a control chart might have been a better way of assessing the change over time as the MDT 'matured'?	To make the status of the '4-week pilot' of the study clearer in relation to the start of the cooperation of the UMCG with MCL during 1993-1997, we changed 'after a 4 year pilot' to 'after an informal cooperation period of 4 years'. The 4-week pilot of the study was carried out 9 months before study start.	In the introduction, page 4, line 98, we changed 'after a pilot of 4 years' to 'after an informal cooperation period of 4 years'; we added in line 99: 'formal'. We added in design, page 6, line 158: In a '4-week' pilot study for the quantitative part of the study, 'carried out 9 months before study start,' We added in a sentence in the discussion, page 19, line 357: Additionally presence of the researcher might have influenced the

no	reviewer comment James Green	reply author	changes in document
			communication between centre and partner. We added in discussion, page 19, lines 359-361: This nuance could also be interpreted as a social desirable answer, possibly due to the long existing collaboration between the centre and the partner before study start.
7	As the amount of change was very low, it would be interesting for the researchers to calculate a) how many patients did NOT need a discussion due to robust clinical guidelines b) an estimate of how many needed to be brought the videoconference joint MDT using their proposals and how much resource this could save (clinicians time etc). Is there any costing for MDT time in the Netherlands? c) If there was a robust way of identifying who needed to be brought for a full discussion – what were the common criteria (even if the small numbers means this is not possible it should be included as a focus for further research)? As the authors are saying that not all cases should be discussed, a guideline on who should be discussed (the category of patient/disease/complexity) would probably be useful going forward. Or it may be easier to say categorically who should NOT be discussed. I realize the authors are saying 'not all new cases' but surely some may necessitate discussion. If so why?	Because of an increase in patients to be presented in the meeting, we are looking for a more efficient meeting, which could be reached not discussing the 'formalities' or 'routine patients' (about 70% of patients); developing an evidence based working method would need more research. This is not a typical VC problem, but a oncology wide guideline item (see also 'routine patient', comment 13, reviewer Joan Prado). At the moment the MDTs struggle with the amount of patients that need to be discussed. Nationwide there is a discussion to select patients for diverse MDTs, however further research would be necessary.	We added to the results, page 11, line 244: About 70% of our patients are 'formalities' or 'routine patients', meaning patients that fit the guidelines (well-defined tumours with limited regional metastases without comorbidity). We added in discussion, page 17, line 308-310: Because of an increase in patients to be presented in the meeting, we were looking for a more efficient meeting, which could be reached not discussing the 'formalities' or 'routine patients' (about 70% of patients); developing an evidence based working method would need more research.
8	Line 355 proposed to discuss more complex cases - supports the finding of CR UK study which also assessed head and neck cancers (see reference above).	Thank you, we refer to the report as additional reference in the introduction. See also comment 5.	We added in the discussion, page 17, line 306: The specialists said that they did not want to stop the video-conferenced MDT, because they appreciate reflecting on diagnostic and treatment plans with trusted expert colleagues.

no	reviewer comment James Green	reply author	changes in document
9	MDTs are thought to save time elsewhere administratively and due to familiarity with the case after presentation. Was there any data to show that staff felt the discussion saved them time/resource elsewhere in the patient care pathway?	The MDT by VC of the centre with partner is not prepared by the other site, the other site being a 'fresh team' that can be seen as 'second opinion'. Interviews were focused on improvement on efficiency for the (additional) video conferenced MDT.	No changes made in the document.
10	Staff seemed to identify other organizational factors that they would like to improve, in the discussion could solutions to these issues be proposed.. The whole team (centre and partner) could have a face to face regularly/annually so that guidelines , cross sites improvement and trials can be proposed, discussed and implemented ,	Thank you for your suggestion. The so called 'working group Northern Head and Neck Cancer' (centre and partner) meets face-to-face 3 times a year to discuss care pathway management issues like governance aspects (such as efficiency of the care pathway, human resources, which quality indicators), changed Dutch guideline or international TNM-classification implementation and research issues. These meetings are seen as 'social cohesion' (see also advice offered to other teams that want to introduce VC in the discussion, page 20).	We added in introduction a sentence, page 4, line 106: The teams of centre and partner meet face-to-face three times a year, where governance, guidelines and research projects are discussed. We added to the advice in the discussion, page 20, line 394: ', which serves cohesion (management meetings on governance, guideline developments and research projects are ideal for this purpose).'
11	There could be more discussion of other MDT models instead of video conferencing (perhaps a virtual model, if the number of cases are so small, the cases could be perhaps put up on a website 1 week before and specialists comment on the best care) that could provide a solution to the resources needed to run this MDT.	Thank you for your interesting opinion on MDT models that involve ICT in different ways. The clinical problems of the H&N are not suitable for web-discussions, because the different oncology treatment options are almost equal in effectiveness and are discussed in the light of morbidity, 'retention of functionality for the patient' and the wishes of the patient (shared decision making). Interaction between the different disciplines is impossible off-line,	No changes made in the document.

no	reviewer comment James Green	reply author	changes in document
		they assess images in the light of their treatment modalities. On top of that, specialists from different hospitals do not have access to each other's databases, because of Dutch privacy law. In our discussion, lines 370-373 (Other studiespatients) we refer to several studies on video-conferenced MDT with a slightly different approach.	
12	Who else is present in the Head and neck MDT as in the UK CNSs and therapists are present, is this the case in the Netherlands? They often think their time is saved later (see above) as they are familiar with the case. Perhaps stating the team that makes up the MDT in each site, under standard guidance/minimum standard expected would be useful. If they are present why were only doctors interviewed?	Thank you for your interest in our organization, although the local or site MDT is not part of our study. As stated before, the video conferenced MDT is an additional MDT between centre and partner, required by the Dutch government. Diagnostic and treatment plans are assessed by the local MDT. A local MDT would consist of specialists of ENT, OMS, MO and RT, the nurse practitioner of ENT and OMS, radiologist, pathologist, dietician, special dental care and medical social worker. We described this in detail in our first paper on evaluation of change in the care pathway for Head and Neck Cancer patients (Reference: 'Multidisciplinary first-day consultation reduces throughput times for head-and-neck patients': http://dx.doi.org/10.1186/s12913-018-3637-1.)	We added text in introduction, page 4, line 105: The centre's MDT regarding diagnostics and treatment involves more than 9 disciplines (details presented elsewhere).
13	Could the authors please state on which sites the 6 specialists worked who were interviewed in the qualitative aspect of the study. Were 3 individuals at the centre and 3 at the partner site? Could their specialities be given and had all finished training i.e. were they all qualified staff	With the changes in the design to reflect the quantitative and the qualitative research elements, we also pointed out how the specialists for the interviews were selected. From both sites 3 specialists were selected, form	We added in design, page 8, line 185: that attended the meetings most frequently.

no	reviewer comment James Green	reply author	changes in document
		OMS-ENT- and RT-department. This latter part was already described in the design.	
14	There appeared to be medical trainees presenting cases, why was this, were people on leave? Would one expect fully trained staff at each site in the model/operating guidance to be doing so ?	The ENT specialist is a fully trained ENT specialist, training for oncology; we will remove the word 'fellow' in the results and the discussion.	Changed in results, page 11, line 232: The centre's ENT department was represented in most meetings by 'an ENT-specialist' training to be a head-and-neck 'oncology' surgeon. Changed in discussion, page 18, line 342: fellow to colleague.
15	I couldn't understand why in table 2 Gender was included as a mean (with SD) when absolute numbers might have been more useful	Subheading of table 2 were not clear enough, it says for gender absolute number (n) and %; the same for tumour localization.	Changed appearance of table 2: subheadings were made clearer with double lining.
16	Attendance seems to be an issue and if full attendance had occurred at the centre then 4 recommendations (proposed by the partner site) might have been removed (leaving 1%) - is cross covering for leave etc a planned function of the centre/partner MDT model, if so can this fact be including in the discussion/introduction. As if so the 2% may not necessary be due to 'change' but due to cross-covering a speciality by one site when others are away.	Thank you for your thoughts on recommendations versus team completeness. It is an interesting aspect. We were describing our results 'as faithfully as possible', but perhaps we were somewhat unclear in our result description. Just like in the site's 'face-to-face MDT' it is sometime difficult to have every discipline present, due to other obligations and sickness. Facts: of the 3 times the centre is not complete, 2 recommendations were given from partner to centre, and 1 was from (not complete team) centre to partner. It is speculative to say that if the centre team would have been complete no recommendations	We added in the results, page 11, line 230: On those occasions one of the other specialisms would present the cases, for example OMS for ENT.

no	reviewer comment James Green	reply author	changes in document
		would have been received. It is however interesting for further research in the future.	
17	346 enjoy team working - There was little mention of other valuable aspects that video conferenced MDT / access to supra-specialist advice can bring: Education of teams/juniors, discussion of whether patients might be eligible for Research trials etc. Where any of these other aspects brought up by the interviewees as important aspects of the meeting? where they asked this? It would perhaps be useful to explain these other positive aspects of MDTs to readers who may be unfamiliar with MDTs. Just meeting over videoconferencing is thought to be suboptimal for team working. Moving from a face to face meeting in one room, to videoconferencing it would be sensible to put more social/team interaction in place to facilitate improvements and development of services. This is often unappreciated so a paragraph about this would be useful.	Thank you for sharing your knowledge on MDT use. As pointed out the video conferenced MDT is an additional MDT as part of the collaboration agreement and is required by the Dutch Healthcare Inspectorate. The team was asked to name improvements and to reflect on stopping the video conferenced MDT. They all did not want to stop (value in complex cases having a 'second opinion' with fresh team), but were looking for more efficiency. In the video conference MDTs there are no juniors or researchers. Thank you, we agree, we added a sentence to the introduction. See comment 10.	We added in the discussion, page 17, line 306: The specialists said they did not want to stop the video-conferenced MDT, because they appreciate reflecting on diagnostic and treatment plans with trusted expert colleagues.
18	In many parts of the world the MDT meeting has a dual role; a governance aspect and to facilitate evidence based decision making on the options of care for patient discussed.	We added on the description of the collaboration between the centre and the partner that agreements were made on governance aspect, evidence base policies and use of facilities.	We added in the introduction, page 4, line 101: sustainable agreements on governance aspects, evidence based multidisciplinary decision making and use of facilities.

Reviewer: 3: Joan Prades, Institution and Country: Catalan cancer plan, Health department, Catalonia, Spain & University of Barcelona (UB)

In the field of oncology, the head and neck cancer is typically known by the conflictual decision-making process given the existing different therapeutic alternatives and the radical impact of these in terms of short, secondary and secondary effects. On the other hand, the introduction of ICTs in cancer care is a main challenge. Both issues make this article relevant. Nevertheless, I would like to make some remarks.

Authors reply

Dear reviewer, dear Joan, thank you for interesting comments on the diagnostic and treatment challenges for Head and Neck cancer treatment. We agree that ICTs have their benefits, but also have their drawbacks.

We reflected on your comment in the table below.

no	reviewer comment Joan Prades	reply author	changes in document
1	I consider the organisation of the Methodology section quite weak. On one side, it is essential to stress how the qualitative strand is “mixed” with the quantitative strand. Just as an option, I would advise the authors using the classical reference of “Creswell JW and Plano Clark VL. Designing and conducting mixed design research. USA: Sage Publications; 2011” in order to better clarify how you organised the whole study. It is of great importance to make explicit elements such as how both strands relate and were mixed. For instance, was that a convergent or a sequential design? Was the design explorative or explanative? That is, were the qualitative results intended to explain the quantitative ones? And, also, were the results from both strands mixed at the study design stage, or were they mixed at the Results or Discussion stages? This should be introduced in the “design” sub-section.	Thank you for giving us the benefit of your knowledge on design for qualitative research. The study design was sequential, first the quantitative part and later the qualitative part was performed; the interviews were explorative on benefits and drawbacks of the VC. But also on possible improvements. We did not ask participants to reflect on the amount of recommendations.	Method was changed on pages 5-9 to reflect the workflow of the study in the quantitative and qualitative part; we added a subheading in design, page 9, line 201: Thematic analysis.
2	A second element that can be stressed is the lack of order in the presentation of both strands. From page 5 to page 8 quantitative and qualitative elements (patient data, qualitative analysis, sample size...) are shown without splitting the quantitative study from the qualitative study. After giving more details about the study design, I would separate the methodological aspects of the quantitative assessment (or evaluation or sub-study) from the qualitative ones.	We agree with the reviewer and organized the design into two parts in two a quantitative and qualitative part.	Method was changed on pages 5-9 to reflect the workflow of the study in the quantitative and qualitative part.
3	Other key points are the following. It was not explained how the interviewees were selected. It seems obvious but a comment could be introduced as long as the sample of informants is quite small and there may be a risk for respondents’ biases such as “social desirability”. Just as an opinion, I do not consider very relevant the distinction between “recommendation” and “suggestion” (p 17, line 300), being the latter rather “induced”. The point is whether the clinical discussion impacted the clinical management of patients, ultimately.	The presence of specialists were analysed, those specialists that attended the most VC’s were selected for the interviews. We agree with the reviewer, that the wording in English might not differ much (recommendation, advice, suggestion). However in our research design we distinguished ‘recommendations as agreed upon’ during the study (that were registered) and	We added in design, page 8, line 185, ‘that attended the meetings most frequently,’.

no	reviewer comment Joan Prades	reply author	changes in document
		'collegial discussions with suggestions' (that were not registered). We think that further research would be interesting on this point.	No changes made in the document.
4	I'm not totally sure of these two points—page 3, line 63 and 66—being strengths of the study. They seem more comments about the results. I think they have nothing to do with the design of the study or the design used.	We agree, this was also pointed out by the editor of BMJ Open. We changed the 'strengths and limitations' section.	Changed strength and limitations, page 3, lines 70-78.
5	Likewise, another key limitation should be added: the results of the study can be biased due to the 4-year pilot carried out previously. This is described by the authors. It could be considered a limitation in view of the low percentages of change in the recommendations.	We are sorry for the confusion, 1993-1997 was not a pilot study, but an 'informal collaboration', before the formal relation as 'preferred partner and centre' collaboration contract was signed. To clarify and make the distinction between this early 'informal collaboration' and the 4-week pilot run in 2016 - about 9 months before study start -, we changed text in the introduction and design. We did not add a limitation on the pilot part of the study.	We changed the text in the introduction, page 4, line 98, to: In 1997, after an informal collaboration period of 4 years, the Medical Centre Leeuwarden became the formal preferred partner. We added in design, page 6, line 158: 4-week pilot study.
6	I would like to acknowledge the work done and the commitment to transparency in showing the coding tree evaluation. It has been really interesting reading the codes' description.	Thank you.	
7	In page 5, line 131, it is explained that patient data were included but table 2 is not mentioned.	Thank you for your thoroughness, we agree.	We added in design, page 6, line 151: (Table 2).
8	Also, in page 6, it is stressed why patients were not involved. If they were not the target, then is not necessary to include a specific sub-section justifying it.	In this case the reference to patient involvement is not very important, but it is a requirement of BMJ Open, besides that it is the policy of the UMCG to declare if and how patients were considered to be involved in designing research. We clarified that the patients were not the main purpose of	No changes made in the document.

no	reviewer comment Joan Prades	reply author	changes in document
		the study, but could benefit from the outcome.	
9	Finally, we do not know the specific type of qualitative analysis performed. Considering the codification process and the inductive way used to create contents, I presume that it corresponds to the thematic analysis .	Thank you for sharing your knowledge on qualitative research design and taking the time to understand our study. We agree and therefore added this to the design.	We added in design in the qualitative part: - a subheading, page 9, line 201 'Thematic analysis' and - a sentence, page 9, line 206: using a thematic analysis approach with main themes recommendations, added value, collaboration and planning.
10	Concerning the Abstract section, I find it difficult to understand therein the meaning of "Collaborating contract" without reading the whole paper (page 2, line 37). I would rephrase it by saying "as part of a regional health policy rule" or something like that.	We agree that in our introduction the collaboration contract is not mentioned, but is called National = DCHI-policy.	We changed the abstract, page 2, line 39: 'Collaborating contract' was removed, referring only to 'a Dutch health policy rule'.
11	Also, it is mentioned in the "objectives" that this is a mixed design study while the "Design"/Methodology section do not contain this information. This should be changed.	Changed set up of design to reflect the qualitative part followed by the quantitative part.	We changed the design, pages 5-9.
12	It is used the word "policy" to mean that you have a treatment plan for a patient (e.g. line 306). Personally, I would not use in this paper the word policy in that sense.	Thank you for your help in improving our English for a better understanding of the difference between policy, medical procedures and treatment plan.	We changed Table 3, page 12; Table 4, page 14; in results, page 16, line 266, in discussion, page 17, line 317.

no	reviewer comment Joan Prades	reply author	changes in document
13	Although this comment goes probably beyond the scope of a reviewer, I wonder about the category “routine patient” mentioned on page 19, line 354. From my experience, for some physicians a routine patient is someone with a low degree of cancer-related clinical complexity, but this is excluding patients with some comorbidity, patient preferences which are not those of clinicians, etc. Reaching a minimum consensus on the concept of “routine patient” could be interesting. It might help operationalising the avoidance of discussions on these patients (and the feeling of wasted time) while there are no patients who deserve to be discussed by both teams and instead they are not.	We consider in an ‘informal definition’ routine patients to be patients that fit the guidelines (well-defined tumours with limited regional metastases without comorbidity). To proof that the new definition and working method would be effective it would require more research (see also comment 5, reviewer James Green).	We added to the results, page 11, line 244: About 70% of case were ‘formalities’ or ‘routine patients’, meaning patients that fitting the guidelines (well-defined tumours with limited regional metastases and without comorbidity). We added in the discussion, page 17, line 308-310: Because of an increase in patients to be presented in the meeting, we were looking for a more efficient meeting, which could be reached not discussing the ‘formalities’ or ‘routine patients’ (about 70% of patients); developing an evidence based working method would need more research.
14	The idea of efficiency is not included in the text. The results of this study are coherent with the idea of providing high-quality care (by promoting horizontal integration between teams from different hospitals) while making this change sustainable. Just an option, this could be included in the conclusions.	Thank you for your suggestion.	We added to the conclusion, page 21, line 405: This leads to a more efficient use of resources and work force.

VERSION 2 – REVIEW

REVIEWER	Joan Prades Catalan Cancer Plan & University of Barcelona, Spain
REVIEW RETURNED	09-Oct-2019
GENERAL COMMENTS	I carefully reviewed the new version of this paper. I'm fully satisfied with the changes made by the authors. They clarified the methodology regarding the mixed methods approach, improved the terminology (e.g. policy, routine patient) and introduced key ideas (e.g. potential bias of social undesirability, efficiency).

	Considering the other reviewer's work and the answers given by the authors, this is to me a consistent paper worth to be published in your Journal.
--	---